# The substrate specificity switch FlhB assembles onto the export gate to regulate type three secretion

Lucas Kuhlen[1,2,6], Steven Johnson [1], Andreas Zeitler[3], Sandra Bäurle[3,7], Justin C. Deme [1,4], Joseph J. E. Caesar[1,4], Rebecca Debo[3,8], Joseph Fisher[1], Samuel Wagner [3,5] & Susan M. Lea [1,4✉]

Protein secretion through type-three secretion systems (T3SS) is critical for motility and virulence of many bacteria. Proteins are transported through an export gate containing three proteins (FliPQR in flagella, SctRST in virulence systems). A fourth essential T3SS protein (FlhB/SctU) functions to "switch" secretion substrate specificity once the growing hook/ needle reach their determined length. Here, we present the cryo-electron microscopy structure of an export gate containing the switch protein from a *Vibrio* flagellar system at 3.2 Å resolution. The structure reveals that FlhB/SctU extends the helical export gate with its four predicted transmembrane helices wrapped around FliPQR/SctRST. The unusual topology of the FlhB/SctU helices creates a loop wrapped around the bottom of the closed export gate. Structure-informed mutagenesis suggests that this loop is critical in gating secretion and we propose that a series of conformational changes in the T3SS trigger opening of the gate through interactions between FlhB/SctU and FliPQR/SctRST.

[1] Sir William Dunn School of Pathology, University of Oxford, Oxford OX13RE, UK. [2] Department of Chemistry, University of Oxford, Oxford, UK. [3] Interfaculty Institute of Microbiology and Infection Medicine (IMIT), University of Tübingen, Elfriede-Aulhorn-Strasse 6, Tübingen 72076, Germany. [4] Central Oxford Structural Microscopy and Imaging Centre, University of Oxford, Oxford OX1 3RE, UK. [5] German Center for Infection Research (DZIF), Partner-Site Tübingen, Elfriede-Aulhorn-Strasse 6, Tübingen 72076, Germany. [6] Present address: Department of Infectious Disease, Imperial College London, London, UK. [7] Present address: Department of Thoracic and Cardiovascular Surgery, University Hospital Tübingen, Tübingen 72076, Germany. [8] Present address: Centre for Applied Geosciences (ZAG), Department of Geosciences, University of Tübingen, Tübingen 72076, Germany. ✉email: susan.lea@path.ox.ac.uk

Type-three secretion is a mechanism of bacterial protein secretion across both inner and outer bacterial membranes. It is found in the virulence-associated injectisome (vT3SS), a molecular syringe, and the bacterial flagellum (fT3SS), a motility organelle[1]. Both families contribute in significant ways to bacterial pathogenesis. vT3SS facilitate secretion not only across the bacterial envelope but also insert translocon proteins at the tip of the needle into the eukaryotic host plasma membrane, allowing direct injection of virulence factors in the host cytoplasm. The fT3SS is responsible for construction of the flagellar filament in both Gram-negative and Gram-positive bacteria, and hence imparts pathogenicity[2], e.g., via the ability to swim towards favourable environments or sense environmental conditions[3].

Type-three secretion system (T3SS) are multi-megadalton protein complexes that are capable of bridging from the bacterial cytoplasm to the extracellular space. At the core of the secretion system is the highly conserved export apparatus (EA)[4,5], which is made up of five predicted transmembrane (TM) proteins (SctR, SctS, SctT, SctU and SctV in the vT3SS; FliP, FliQ, FliR, FlhB and FlhA in the fT3SS). FlhA/SctV has been shown to form a nonameric ring[6-8], consisting of a large cytoplasmic domain situated below a hydrophobic domain predicted to contain 72 helices. This structure was proposed to surround an "export gate" through which substrates would enter the secretion pathway. This export gate is constructed from the other four EA proteins and was predicted to lie in the inner membrane. However, our recently determined structures of the *Salmonella enterica* serovar *Typhimurium* $FliP_5Q_4R_1$ and the *Shigella flexneri* $SctR_5S_4T_1$ complexes[9,10] demonstrated that the export gate is actually embedded within the proteinaceous core of the T3SS basal body, placing it above the inner membrane. Furthermore, the helical structure of the export gate makes it likely that it is responsible for nucleating the helical filaments that assemble above it[11]. Interestingly, the EA has also recently been proposed to facilitate inward transport across the inner membrane associated with nanotubes[12,13]. The final component of the EA, FlhB/SctU, has long been known to be essential for all T3SS-mediated protein secretion. In addition, FlhB/SctU has a major role in switching the specificity of secretion substrates, mediating the transition from the early components necessary to build the flagellar hook in fT3SS and injectisome needle in vT3SS, to the later subunits required for flagellar filament or injectisome translocon assembly. The FlhB/SctU family of proteins all contain an N-terminal hydrophobic sequence that is predicted to form four TM helices ($FlhB_{TM}$) and a smaller cytoplasmic C-terminal domain ($FlhB_C$). Crystal structures of the FlhB/SctU cytoplasmic domain from a range of species and systems[14-16] demonstrated a compact fold with an unusual autocatalytic cleavage site in a conserved NPTH sequence. Cleavage between the Asn and Pro residues, splitting $FlhB_C$ into $FlhB_{CN}$ and $FlhB_{CC}$, is required for the switching event to occur and a variety of mechanisms have been proposed to explain the need for this unusual mechanism[17].

Little was known about the predicted TM portion of FlhB/SctU. Co-evolution analysis and molecular modelling led to suggestions that it forms a four-helix bundle in the membrane[18], whereas crosslinks[19] and partial co-purification of FlhB with FliPQR were consistent with FlhB/SctU interacting with the export gate via a conserved site on the $FliP_5Q_4R_1$ complex[9]. However, given the inaccuracy of the TM predictions for the other export gate components revealed by the FliPQR structure, we sought to determine the molecular basis of the interaction of FlhB with FliPQR. Here we present the structure of the TM region of FlhB bound to the FliPQR complex, in addition to the structures of the FliPQR complex from *Vibrio mimicus* and *Pseudomonas savastanoi*. The structure reveals a unique topology that presents a loop that wraps around the base of the closed export gate. Mutagenesis studies confirm the crucial role played by the FlhB loop in the export process and suggest potential mechanisms of regulation of opening of the assembly.

## Results

**Conservation of the FliPQR structure**. Our previously determined structures of *S. typhimurium* FliPQR[9] and the vT3SS homologue SctRST from *S. flexneri*[10] demonstrated that the stoichiometry of the core structure ($FliP_5Q_4R_1$) is conserved between virulence and flagellar systems. However, classification of the SctRST data revealed variable occupancy of the SctS component (up to a maximum of four copies), consistent with our earlier native mass spectrometry measurements[9]. Furthermore, our native mass spectrometry had also demonstrated that a small proportion of the *P. savastanoi* FliPQR complex contained five copies of FliQ, with the predicted fifth FliQ-binding site beginning to encroach on the predicted FlhB interaction site. To further analyse the structural conservation and stoichiometry of the EA core FliPQR, we chose the homologous complexes from the fT3SS of two other bacterial species for structural studies: the *V. mimicus* polar flagellum FliPQR complex, which has a longer FliP sequence including an N-terminal domain conserved in the *Vibrionales* order (Supplementary Fig. 1), and the *P. savastanoi* FliPQR complex, which is a mixture of $FliP_5Q_5R_1$ and $FliP_5Q_4R_1$ complexes by native mass spectrometry[9]. We determined the structures of both complexes using single-particle cryo-electron microscopy (cryo-EM) analysis to 4.1 Å and 3.5 Å, respectively (Fig. 1a, Table 1, Supplementary Fig. 2 and Supplementary Fig. 3). Both structures are highly similar to *S. typhimurium* FliPQR[9] (root-mean-square deviation (RMSD) = 1.6 Å over all chains) and *S. flexneri* SctRST (*V. mimicus* FliPQR and SctRST RMSD = 1.9 Å and *P. savastanoi* FliPQR and SctRST RMSD = 2.3 Å)[9,10].

Consistent with our previous native mass spectrometry data, the structure of *P. savastanoi* revealed an additional FliQ subunit in the complex. In the *S. typhimurium* and *V. mimicus* FliPQR structures, there are four FliP–FliQ units, each the structural equivalent of a FliR subunit[9], but the fifth FliP is missing a FliQ. In the *P. savastanoi* structure, $FliQ_5$ binds the remaining FliP subunit in the same way as in the other FliP–FliQ units. This $FliQ_5$ subunit is located close to the site on FliPQR we previously identified as important for interaction with FlhB/SctU[9,19]. We conducted an extensive in-vivo photocrosslinking analysis based on covariance[18] between SctU, SctR, SctS and SctT, which supports a binding site for SctU that involves large parts of helix 2 of SctS and helix 4 of SctT (Fig. 1b, c and Supplementary Fig. 4). Mapping of the residues on the structure of *S. typhimurium* FliPQR and a model of FliPQR containing a fifth FliQ subunit reveals a large binding site in the complex containing four FliQ subunits and a more compact binding site when a fifth FliQ subunit is modelled (Fig. 1d). In this way, $FliQ_5/SctS_5$ might be compatible with FlhB/SctU binding, depending on the unknown structure of the FlhB/SctU TM domain ($FlhB/SctU_{TM}$), but addition of a sixth FliQ/SctS using the same helical parameters would block this site.

**Architecture of the FliPQR–FlhB export gate complex**. We have observed four FliQ subunits in the *S. typhimurium* and *V. mimicus* FliPQR and the *S. flexneri* SctRST structures; however, as we have previously observed FliQ to be sensitive to dissociation by detergent in the purification process[9,10], it was possible that the fifth FliQ is a genuine component of the complex but was lost in the purification of less stable homologues. As the stoichiometry of FliQ has potentially large implications for the placement of FlhB in the system (Fig. 1d), we endeavoured to produce the more physiologically relevant FliPQR–FlhB complex.

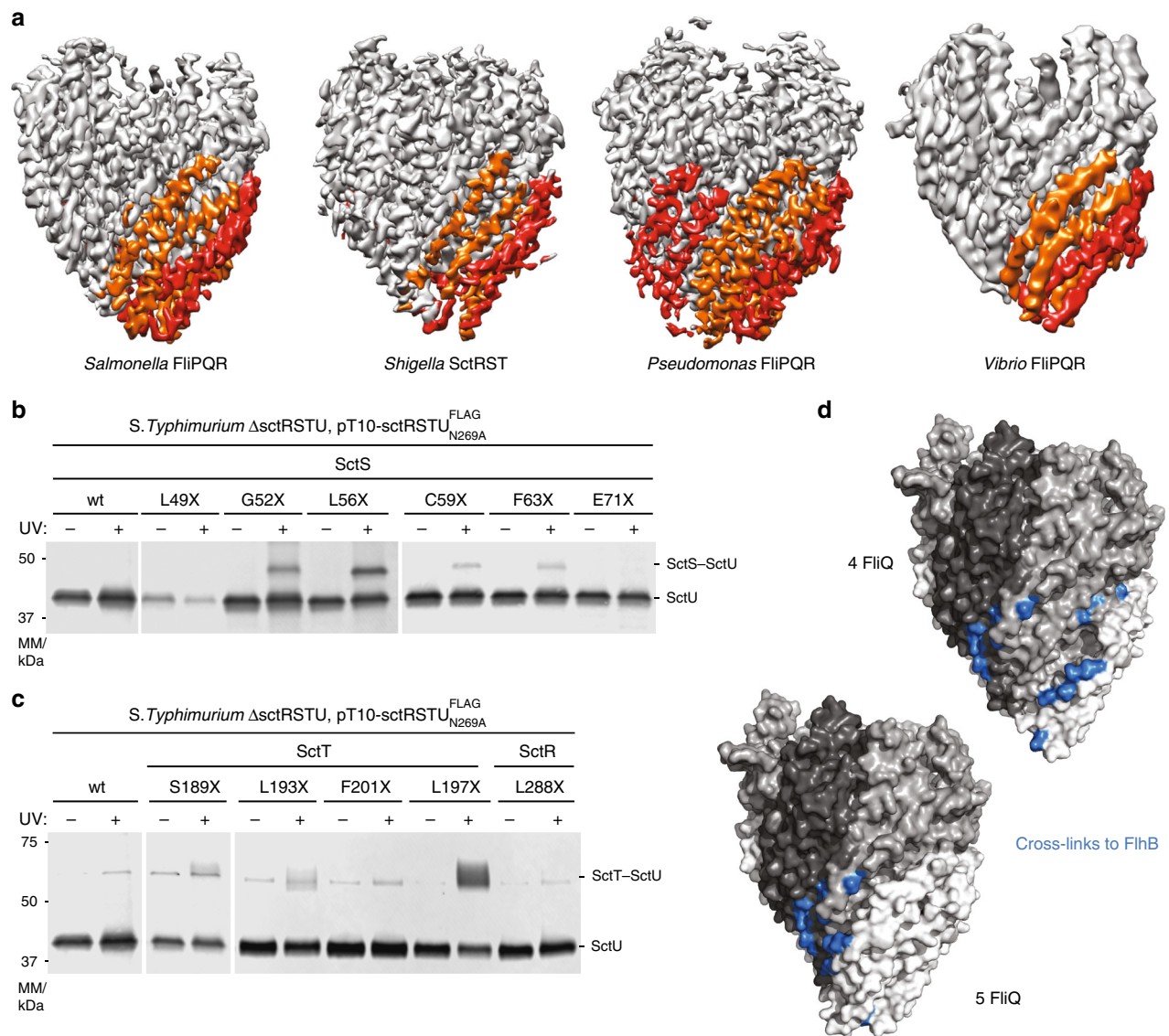

**Fig. 1 Conservation of the structure of the FliPQR export gate in the closed state. a** Cryo-EM volumes calculated in Relion using data from *S. typhimurium* FliPQR (left, EMD-4173), *S. flexneri* SctRST (centre left, SctR$_5$S$_4$T$_1$ class[10] (EMD-4734)), *P. savastanoi* FliPQR (centre right) and *V. mimicus* FliPQR (right). FliQ$_2$ and FliQ$_4$ are coloured orange and FliQ$_1$, FliQ$_3$ and FliQ$_5$ are coloured red. **b** Immunodetection of SctU$^{FLAG}$ on western blottings of SDS-PAGE-separated crude membrane samples of the indicated *S. typhimurium* SctS *p*Bpa mutants (denoted with X). Each sample is shown with and without UV-irradiation to induce photocrosslinking of *p*Bpa to neighbouring interaction partners. **c** As in **b**, but testing interactions to SctU with *p*Bpa in SctT and SctR. **d** Mapping of the confirmed contact points between FliPQR/SctRST and FlhB/SctU, including those previously identified[19] on the structure of FliPQR (*S. typhimurium*) and a model with a fifth FliQ subunit, which is based on the structure of *P. savastanoi* FliPQR.

After extensive screening of detergents, constructs with different placement of affinity tags and sequences from a variety of species for co-expression and co-purification of FlhB with FliPQR, we were able to prepare a monodisperse sample of the complex from *V. mimicus* (Fig. 2a). We analysed this sample by cryo-EM and determined the 3.2 Å structure of the complex (Fig. 2b, Table 1 and Supplementary Fig. 5), revealing a single copy of FlhB added to the previously observed FliP$_5$Q$_4$R$_1$ complex. The FliPQR subcomplex in this structure is very similar to the structure of the FliPQR complex in the absence of FlhB (RMSD = 0.6 Å), whereas FlhB is observed to contain four long helices in the putative TM domain (FlhB$_{TM}$), forming two distinct hairpins that are wrapped around the outside of the FliPQR complex. This extensive interaction surface between FlhB and FliPQR reveals FlhB to be an integral part of the core of the EA instead of an accessory factor. The opened out structure of the

four predicted TM helices of the FlhB$_{TM}$ domain once again highlights the potential dangers in predicting complex structures in the absence of some of the subunits. The soluble, globular, cytoplasmic domain (FlhB$_C$) is not visible, likely due to flexibility in the linker between the two domains. However, to assess whether the disorder was due to the presence of the detergent micelle in our sample, we also imaged FliPQR–FlhB in the amphipol A8-35, perhaps a better mimic of the proteinaceous environment relevant to the assembled T3SS[9]. We did not observe any additional density resulting from ordering of FlhB$_C$ (Supplementary Fig. 6).

Intriguingly, the two helical hairpins of FlhB are joined by a loop (FlhB$_L$) that literally loops around the (closed) entrance of the FliPQR gate (Fig. 2c). Consistent with our previous prediction[9] and crosslinking analysis (Fig. 1), FlhB contacts the site across FliP$_5$ and FliR, and it additionally contacts

**Table 1 Cryo-EM statistics.**

| | FliPQR, *P. savastanoi* (EMD-10095) (PDB 6S3R) | FliPQR, *V. mimicus* (EMD-10096) (PDB 6S3S) | FliPQR–FlhB, V. mimicus (EMD-10093) (PDB 6S3L) | FliPQR–FlhB amphipol, *V. mimicus* (EMD-10653) |
|---|---|---|---|---|
| Data collection and processing | | | | |
| Magnification | 165,000 | 165,000 | 165,000 | 165,000 |
| Voltage (kV) | 300 | 300 | 300 | 300 |
| Electron exposure (e⁻/Å²) | 48 | 48 | 48 | 48 |
| Defocus range (μm) | 0.5–4 | 0.5–4 | 0.5–4 | 0.5–4 |
| Pixel size (Å) | 0.822 | 0.822 | 0.822 | 0.822 |
| Symmetry imposed | C1 | C1 | C1 | C1 |
| Initial particle images (no.) | 503,177 | 1,050,955 | 1,386,230 | 677,403 |
| Final particle images (no.) | 97,987 | 243,489 | 162,408 | 137,136 |
| Map resolution (Å) | 3.5 | 4.1 | 3.2 | 4.0 |
| FSC threshold | 0.143 | 0.143 | 0.143 | 0.143 |
| Refinement | | | | |
| Initial model used | EMDB-4173 | EMDB-4173 | EMDB-4173 | |
| Model resolution (Å) | 3.5 | 4.1 | 3.2 | |
| FSC threshold | 0.143 | 0.143 | 0.143 | |
| Map sharpening $B$ factor (Å²) | −101 | −214 | –97 | |
| Model composition | | | | |
| Non-hydrogen atoms | 12,855 | 12,321 | 13,849 | |
| Protein residues | 1669 | 1569 | 1768 | |
| Ligands | 0 | 0 | 0 | |
| B factors (Å²) | | | | |
| Protein | 36.94 | 101.85 | 43.55 | |
| Ligand | NA | NA | NA | |
| R.m.s. deviations | | | | |
| Bond lengths (Å) | 0.0058 | 0.0066 | 0.01 | |
| Bond angles (°) | 0.88 | 0.87 | 0.93 | |
| Validation | | | | |
| MolProbity score | 2.54 | 2.25 | 2.45 | |
| Clashscore | 18 | 18.28 | 15.56 | |
| Poor rotamers (%) | 2.22 | 0.07 | 2.72 | |
| Ramachandran plot | | | | |
| Favoured (%) | 90.82 | 91.86 | 93.64 | |
| Allowed (%) | 8.39 | 7.76 | 5.84 | |
| Disallowed (%) | 0.79 | 0.39 | 0.52 | |

cross-linkable residues in the FliP₄ subunit (Fig. 2d). Interestingly, both termini of FlhB$_{TM}$ are cytoplasmic, whereas the C terminus of FliR is periplasmic. Perplexingly, the observation that an engineered fusion of FliR–FlhB in *Salmonella*[20] weakly complements a double *fliR/flhB* knockout, in conjunction with the existence of a natural FliR–FlhB fusion in *Clostridium*, led to previous suggestions that the C terminus of FliR and the N terminus of FlhB are either both cytoplasmic or both periplasmic. In light of our structure, we would either have to accept that the N terminus of FlhB (residues 1–28 that are not observed in our structure) can wrap over the surface of the FliPQR–FlhB complex to reach towards the C terminus of FliR, or that the low level of activity seen in the engineered *Samonella* fusion *(20)* and activity in the native *Clostridial* species relies on proteolytic separation of the two proteins.

As previously predicted[9], hydrophobic cavities between FliP and FliQ, in addition to lateral cavities between the FlhB hairpins and the FlhB/FliQ interface, are observed to contain densities consistent with lipid or detergent molecules, although these could not be modelled unambiguously in the current volume (Supplementary Fig. 7).

**Structure of the hydrophobic domain of FlhB.** The density corresponding to FlhB was of sufficient quality to build an atomic model of the structure using only sequence information (Fig. 3a and Supplementary Fig. 5). The topology of FlhB$_{TM}$ is unusual; the helices 1 and 4 neighbour each other in the middle of the structure, whereas helix 2 and 3 flank the central pair on either side (Fig. 3b, c). To further validate the topology of FlhB, we compared our model with contacts derived from evolutionary co-variation (Fig. 3d and Supplementary Fig. 8). This shows strong contacts between helices 1 and 2, 3 and 4, and 1 and 4 but an

absence of contacts between helices 2 and 3, which is inconsistent with a helical bundle but consistent with our more extended and topologically unusual structure.

Despite observing up to five FliQ subunits in different FliPQR structures, there are only four FliQ molecules in this structure. In fact, the hairpin composed of FlhB helices 1 and 2 is bound to the site occupied by FliQ₅ in our *P. savastanoi* FliPQR structure, packing on FliP₅, whereas helices 3 and 4 pack on FliR. Thus, the presence of FliQ₅ would block binding of FlhB (Fig. 3e), suggesting that the fifth FliQ binds to the complex in a non-native manner due to the absence of FlhB in the overexpression system. This superposition of FlhB and FliQ₅ also reveals that the hairpin of helices 1 and 2 of FlhB adopts a very similar structure to FliQ, despite the fact they are topologically distinct, with helix 2 of FlhB being structurally equivalent to helix 1 of FliQ and vice versa, i.e., the directionality of the hairpin is reversed along the long axis. Modelling FliQ₅ and FliQ₆ using the same helical parameters by which FliQ₁ to FliQ₄ are related reveals that FlhB continues the spiral of FliQ subunits and even helices 3 and 4 follow the same parameters (Fig. 3e) despite not interacting with a FliP subunit. Given the very different topologies of the two proteins, the level to which FlhB helices 1 and 2, and FliQ superpose is surprising. An evolutionary relationship between FlhB and FliQ is unlikely due to the topology differences, suggesting that the similarity of the structures is a result of convergent evolution and the need to form this helical assembly.

**An extended loop between helices is essential for secretion.** The unusual topology of FlhB$_{TM}$ means that a long loop, FlhB$_L$, between helix 2 and 3 (residues 110–139) connects the two hairpins of the structure. Most unexpectedly, this loop, which contains the most highly conserved residues within FlhB

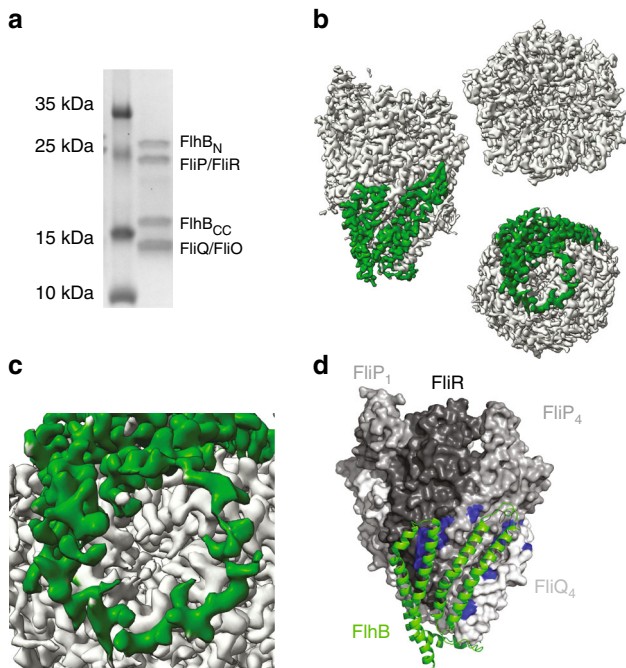

**Fig. 2 Architecture of the FliPQR–FlhB export gate complex. a** SDS-PAGE gel of the *V. mimicus* FliPQR–FlhB complex. **b** Cryo-EM density map of the FliPQR–FlhB complex with the density corresponding to FlhB coloured in green. **c** Zoom of the FlhB loop at the bottom of the complex. **d** Model of FliPQR (surface) and FlhB (cartoon, green) with residues in FliP/SctR and FliR/SctT crosslinking to FlhB/SctU highlighted in blue as in Fig. 1d.

(Supplementary Fig. 9), is seen to wrap around the base of the PQR complex, contacting each of the FliQ subunits in turn and inserting conserved hydrophobic residues into the cavities between the FliQs (Fig. 3f). The loop structure also reveals how a single FliQ residue can co-evolve with multiple FlhB$_L$ residues. Although FlhB$_L$ in isolation does not further constrict the base of the already closed PQR complex, the aperture does become significantly smaller when taking into account the poorly resolved termini of FlhB$_{TM}$ (Supplementary Fig. 10). Therefore, FlhB$_L$ could contribute to export gate closure via trapping of the FlhB N terminus and the linker connecting FlhB$_{TM}$ to FlhB$_C$, in the direct line of the export pathway or by pinning the FliQ subunits closed. A mutation in the FlhB N terminus had been reported[21] to act as a ΔfliHI bypass mutant (the ATPase and its regulator), presumed to be involved in controlling the opening of the export channel. In the FliPQR–FlhB complex of *V. mimicus*, the equivalent residue (FlhB$_{P28}$ in *S. typhimurium* and FlhB$_{A28}$ in *V. mimicus*) locates very close to the pore entrance (Supplementary Fig. 10). In the *S. typhimurium* SctRSTU complex, the corresponding residue strongly photocrosslinks to SctS (Supplementary Fig. 4), supporting the notion that SctU mediates gating of the EA core complex.

We decided to further probe the function of FlhB$_L$ using mutagenesis in the motile *Escherichia coli* strain W (Fig. 4a and Supplementary Fig. 11). We have used *V. mimicus* residue numbering throughout (Supplementary Table 1). Given that opening of the FliPQR–FlhB aperture would require a conformational change in FlhB$_L$, we hypothesized two mechanisms for such conformational changes. FlhB$_L$ could either move away from the entrance to the gate through a hinging motion like a lid or it could extend into a structure with less secondary structure to stay in contact with the binding sites on the opening FliQ subunits, reminiscent of a sphincter.

Mutations in either the conserved hydrophobic residues of FlhB$_L$ that insert between the FliQ subunits (Fig. 3f) or the highly conserved loop of FliQ severely reduced motility (Fig. 4a), protein secretion (Supplementary Fig. 11) and formation of flagella (Supplementary Fig. 13) without affecting EA assembly (Supplementary Fig. 14). Although substitution with the bulky, hydrophobic amino acid tryptophan and removal of bulky sidechains only reduced motility, introduction of charged residues completely abolished motility, suggesting that secretion can proceed at lower efficiency when the FliQ–FlhB$_L$ interaction is only reduced rather than completely disrupted as in the aspartate and arginine mutants.

We performed an extensive in-vivo photocrosslinking analysis to validate the interactions and functional relevance of the corresponding SctU$_L$ in the vT3SS-1 of *S. typhimurium*. Although no crosslinks to SctS could be identified with the artificial crosslinking amino acid *p*-benzoyl-phenylalanine (*p*Bpa) introduced into SctU$_L$ itself (Supplementary Fig. 15), numerous crosslinks were identified with *p*Bpa in the lower part of SctS that faces SctU$_L$ (Fig. 4b). Using two-dimensional blue native/SDS-polyacrylamide gel electrophoresis (PAGE), we could show that the crosslink observed with SctS$_{Q43X}$ occurred not only in the SctRSTU assembly intermediate but also in the assembled needle complex (Fig. 4c), adding further support to the idea that the structure of the isolated complex represents the structure of the complex in the full assembly. The observed crosslinks were independent of functional secretion of the vT3SS, indicating that assembly of needle adapter, the inner rod and needle filament does not lead to a conformational change of this part of the SctRSTU complex (Fig. 4b).

Our motility assays (Fig. 4a) could be influenced by multiple factors including growth of the cells, assembly of flagella, chemotaxis and secretion. In addition to assaying secretion of FliC directly in the flagellar strains (Supplementary Fig. 12), we also directly assayed secretion of the early secretion substrate SctP in the *p*Bpa mutants. Strikingly, introduction of *p*Bpa at several positions of SctS led to a strong defect in secretion but not SctS–SctU$_L$ interaction, highlighting the relevance of this site for secretion function of T3SS (Fig. 4d), whereas *p*Bpa substitutions within SctU$_L$ were more functionally neutral. In total, we found a large number of residues at the FliQ/SctS-FlhB$_L$/SctU$_L$ interface that are required for type-three secretion (Fig. 4e).

Strong crosslinking between SctS and SctU even in the assembled needle complex and loss of function in more disruptive mutations in which the interaction between FliQ and FlhB$_L$ is altered through the introduction of charged residues suggest that this interaction is important for activity and FlhB$_L$ is not simply one of the closure points of the complex in assembly intermediates. If this interaction is maintained in the open state of the export gate, FlhB$_L$ would have to adopt a more extended conformation, requiring a minimum number of residues in the loop. Consistent with this idea, deletions of six or more residues in FlhB$_L$ led to loss of motility, while duplication of six residues in FlhB$_L$ was rather better tolerated, leading only to a reduction in motility (Fig. 4a).

We further investigated the function of FlhB$_L$ through more targeted mutations. FlhB$_L$ is a largely extended polypeptide with little secondary structure, but a short stretch at its C terminus is helical. Interestingly, a mutation of a glycine in this α-helix in the FlhB homologue YscU disrupted secretion in the *Yersinia* vT3SS[22]. The equivalent mutation in *E. coli* FlhB, G133D (Supplementary Fig. 11), or mutation of the conserved positively charged residue R136 (Fig. 4a) had little effect assessed at the level of motility, although we cannot rule out more subtle effects on the efficiency with which secretion occurs.

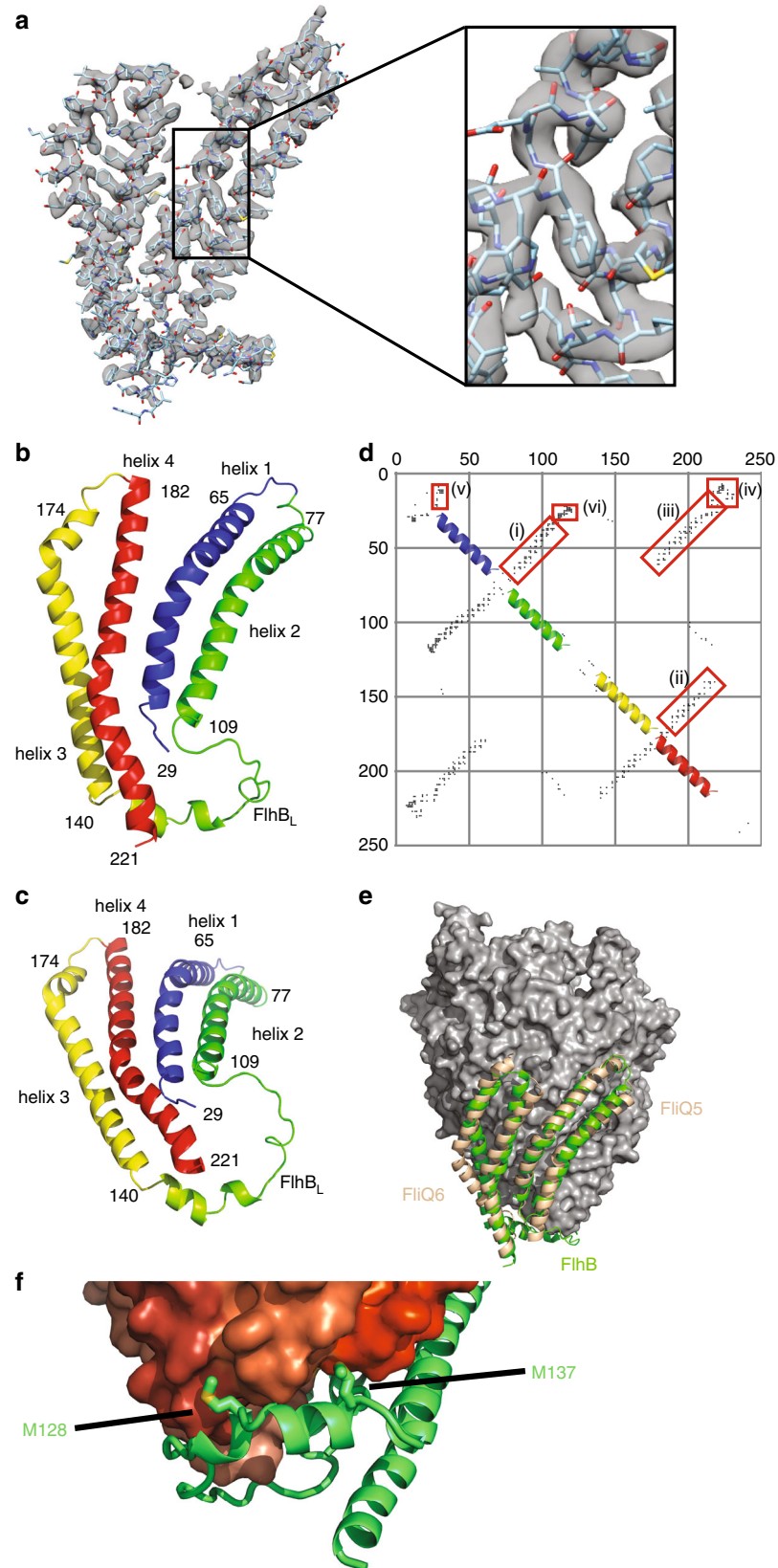

**Fig. 3 Structural analysis of the FlhB hydrophobic domain. a** Quality of the cryo-EM volume corresponding to FlhB. Zoom box shows the fit to density of the model. **b**, **c** Rainbow colouring of the FlhB model with numbers indicating the N and C termini of the four helices. **d** Evolutionary co-variation within FlhB calculated using RaptorX[46]. Only contacts with a probability >0.5 are plotted. Red boxes highlight the interaction between helix 1 and 2 (i), helix 1 and 4 (ii), helix 3 and 4 (iii), the N terminus and FlhB$_{CN}$ (iv), within the N terminus (v) and between FlhB$_L$ and the N terminus (vi). **e** Overlay of FlhB (green) and a modelled FliQ$_5$ and FliQ$_6$ following the same helical parameters as FliQ$_1$ to FliQ$_4$ in *V. mimicus*. **f** Zoomed view of the interaction between the FlhB loop and FliQ, highlighting the intercalation of conserved hydrophobic residues in FlhB between the FliQ subunits.

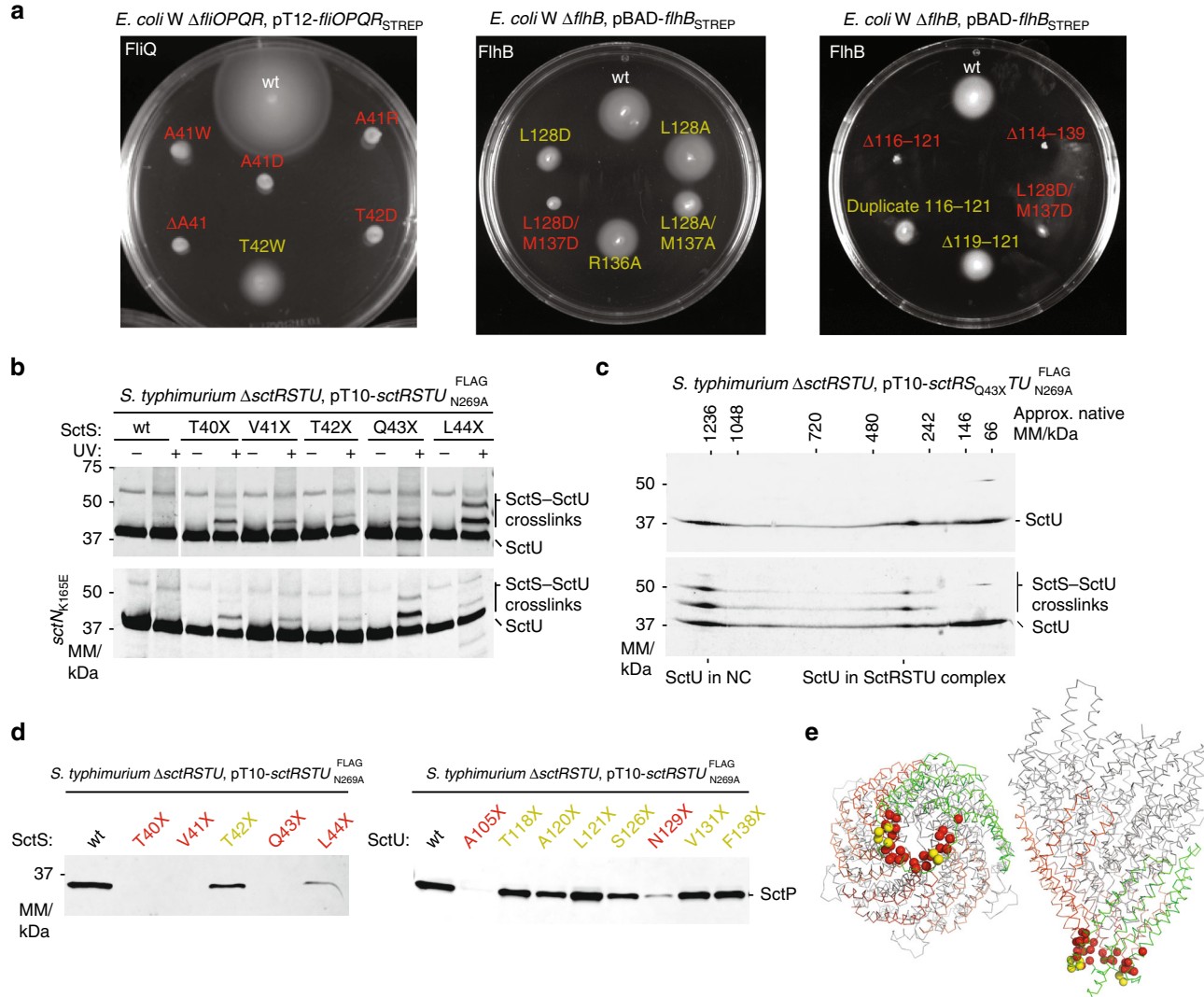

**Fig. 4 Functional analysis of FlhB_L. a** Motility in soft agar of *E. coli* W Δ*fliOPQR* complemented with plasmids expressing *fliOPQR* with the indicated mutations in FliQ (left) and *E. coli* W Δ*flhB* complemented with plasmids expressing FlhB with the indicated mutations (middle and right). **b** Immunodetection of SctU^FLAG on western blottings of SDS-PAGE-separated crude membrane samples of the indicated *S. typhimurium* SctS *p*Bpa mutants (denoted with X). Each sample is shown with and without UV-irradiation to induce photocrosslinking of *p*Bpa to neighbouring interaction partners. Crosslinks between SctS_*p*Bpa and SctU^FLAG are indicated. Crosslinking analysis was performed in the wild type and in an ATP hydrolysis-deficient SctN_K165E mutant that is incapable of type III secretion. **c** Immunodetection of SctU^FLAG on western blottings of 2D blue native/SDS-PAGE-separated crude membrane samples of the indicated *S. typhimurium* SctS *p*Bpa mutant. The sample is shown with and without UV-irradiation to induce photocrosslinking of *p*Bpa to neighbouring interaction partners. Crosslinks between SctS_*p*Bpa and SctU^FLAG in the SctRSTU assembly intermediate and in the assembled needle complex (NC) are indicated. **d** Immunodetection of the early T3SS substrate SctP on western blottings of SDS-PAGE-separated culture supernatants of the indicated *S. typhimurium* SctS *p*Bpa mutants. **e** Structure of FliPQR–FlhB highlighting mutation sites that impaired motility or secretion in red and mutation sites that had no or only a small effect in yellow.

## Discussion

In this study we show that FlhB_TM is part of the export gate complex together with FliPQR. Two pairs of helices of FlhB bind to FliPR through a structure mimicking the shape of FliQ, despite topological reversal, an example of molecular convergent evolution. The unusual topology of FlhB places helices 2 and 3 apart from each other allowing them to mount a loop, FlhB_L, onto the cytoplasm-facing surface of the export gate. Although the way in which FlhB_L wraps around the closed pore suggests a role in maintaining the closed state, our structures of FliPQR/SctRST in the absence of FlhB/SctU are also closed[9,10], as is the complex in the context of the assembled T3SS[23]. This suggests FlhB may be involved in opening of the gate rather than locking it closed, although this would require the linker to be able to extend.

The location and the topology of FlhB_TM place the N terminus, FlhB_CN and FlhB_L in close proximity just underneath the aperture of the gate. Although the resolution of the map is poor in the region of the cytoplasmic face of the complex, it is possible to trace the approximate position of the FlhB N terminus and FlhB_CN (Supplementary Fig. 10). The close association of the N terminus and FlhB_CN is consistent with the strong contacts derived from evolutionary co-variation between the N terminus and the N-terminal part of the cytoplasmic domain (FlhB_CN) (Fig. 3c), and a genetic interaction in *S. typhimurium* between E11 and E230, in the N terminus of FlhB and FlhB_CN, respectively[24]. Furthermore, the direction in which FlhB_CN leaves the export gate implies that, in the context of the assembled T3SS nanomachine, FlhB_C could be located anywhere between the FliPQR–FlhB gate

**a**

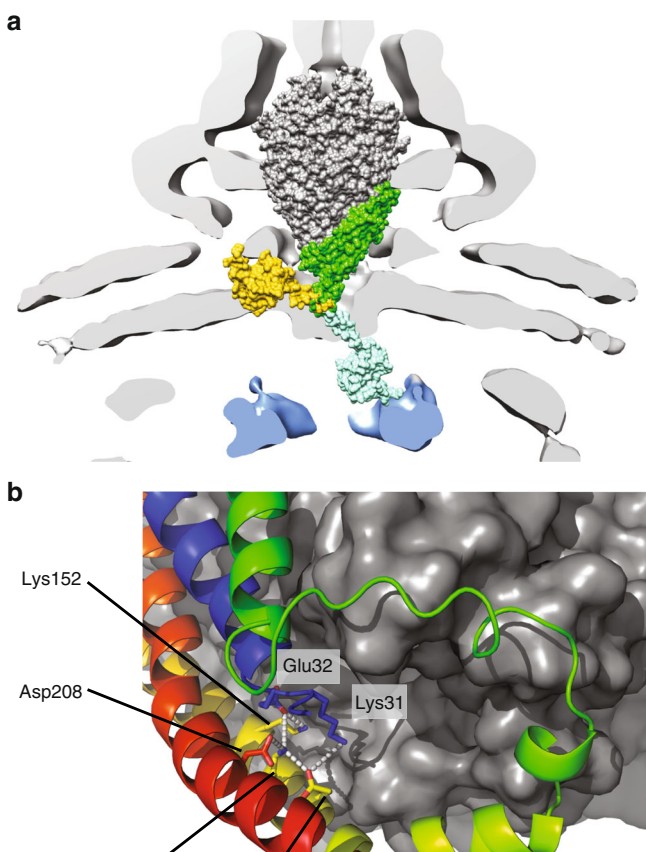

**b**

Lys152

Asp208

Glu32

Lys31

Lys148

Glu145

**Fig. 5 Position of FlhB$_C$ in the complete T3SS. a** Placement of FliPQR–FlhB in a tomographic reconstruction of the *Salmonella* SPI-1 vT3SS (EMD-8544)[47]. FliPQR is shown in grey, FlhB$_{TM}$ in green, two possible positions of FlhB$_C$ (PDB: 3b0z)[15] in yellow and light blue, and the density corresponding to the FlhA homologue is highlighted in blue. **b** Network of salt bridges formed by conserved charged residues in FlhB.

and the nonameric ring of the FlhA cytoplasmic domain below (Fig. 5a). It is conceivable that the conformational changes required for opening of the export gate are propagated via pulling forces imparted on helix 4 of FlhB$_{TM}$, which is linked to the other helices of FlhB$_{TM}$ via a conserved network of buried charged residues (Fig. 5b). Such forces could initiate in the cytoplasmic domain of FlhA, which binds to substrate–chaperone complexes and has been demonstrated to exist in multiple conformations[25,26] marking it as the only component of the EA observed in multiple conformational states to date. Interestingly, this network of charged residues in FlhB includes D208. The D208A mutation disrupts motility, but this defect can be rescued by overexpression of FlhA[27], reinforcing the functional link between the charge network in FlhB and FlhA. As FlhB$_C$ is thought to interact with substrates just before they pass through the export gate[28], it is possible that transition of FlhA between the different conformations pulls on FlhB$_C$, either directly or via substrate, thereby pulling on the FlhB$_{TM}$ network, leading to an opening of the complex. Alternatively, changes in the FlhA$_{TM}$ domain that are triggered via FlhA$_C$ or by the proton-motive force[29] could directly influence the conformation of the FlhB$_{TM}$ domain.

The mechanism of suppressor mutations in the N-terminal residues of FlhB, such as the P28T mutation that rescues motility of

a strain deleted for the ATPase complex (Δ*fliHI*)[21,30], has long been mysterious. Our structure, demonstrating the clustering of the N terminus, FlhB$_{CN}$ and FlhB$_L$ at the cytoplasmic entrance of the gate, suggests that they may rescue function by altering the dynamics of the closure point to facilitate opening of the export gate. This notion is further supported by the interaction we observe of SctU$_{F28pBpa}$ with SctS. This functional link between the ATPase and the gating mechanism, in conjunction with a host of other mutational data in FlhA and FlhB, suggests that cycles of ATP hydrolysis may induce conformational changes in the export gate. As the distance between the export gate and ATPase is large, these conformational changes would presumably be mediated via the FlhA ring that is positioned between them and, has been shown to interact with the FliJ stalk of the ATPase complex[31,32]. As low levels of secretion are possible even in the absence of the ATPase complex or in mutants impaired in ATPase activity[21], low level, spontaneous, opening of the export gate complex must also be possible. Whether the gate subsequently stays open, e.g., due to the continued presence of substrate in the channel[28], or whether it opens and closes as one substrate molecule after the other is injected into the growing filament, as might be expected from the injection-diffusion model of flagellar growth[33], is unknown.

Finally, FlhB/SctU is known to play a key role in substrate switching, an event which requires autocatalytic cleavage of the NPTH sequence in FlhB$_C$/SctU$_C$[16,17,28,34]. Although we do not observe the residues involved in switching in our export gate structure, the fact that we are able to crosslink SctU in fully assembled basal bodies, using the same residues as in the purified complex, suggests that the gating mechanism discussed here is likely applicable regardless of substrate. Clearly, future studies will need to focus on observing gating and switching events.

In summary, our structure of FlhB as part of the export gate complex extends our understanding of the regulation of the T3SS EA and suggests possible mechanisms of export gate opening.

## Methods

**Materials.** Chemicals were from Sigma-Aldrich, unless otherwise specified. The detergents *n*-dodecylmaltoside (DDM) and lauryl maltose neopentyl glycol (LMNG) and the amphipol A8-35 were from Anatrace. *p*BPA was from Bachem. Primers are listed in Supplementary Table 2 and were synthetized by Invitrogen or Eurofins.

**Bacterial strains and plasmids.** Bacterial strains and plasmids used in this study are listed in Supplementary Table 3. Plasmids were generated by Gibson assembly of PCR fragments using the NEBuilder HiFi Master Mix (NEB) or in-vivo assembly[35]. Fragments were created by PCR with the relevant primers using Q5 polymerase (NEB) and genomic DNA templates obtained from DSMZ (*V. mimicus* strain DSM 19130 and *P. savastanoi*, pv. phaseolicola 1448A strain DSM 21482). Gibson assembly and PCR were carried out following the manufacturer's recommendations. *E. coli* W for motility assays was obtained from DSMZ (DSM 1116). Bacterial cultures were supplemented as required with ampicillin (100 μg/mL) or kanamycin (30 μg/mL or 60 μg/mL for large-scale expression in terrific broth (TB). *S. typhimurium* strains were cultured with low aeration at 37 °C in Luria Bertani (LB) broth supplemented with 0.3 M NaCl to induce expression of genes of SPI-1. As required, bacterial cultures were supplemented with tetracycline (12.5 μg/ml), streptomycin (50 μg/ml), chloramphenicol (10 μg/ml), ampicillin (100 μg/ml) or kanamycin (25 μg/ml). Low-copy plasmid-based expression of SctRSTU$^{FLAG}$ was induced by the addition of 500 μM rhamnose to the culture medium.

**Generation of chromosomal deletion mutants.** Electrocompetent *E. coli* W expressing λ Red recombinase from plasmid pKD46 were transformed with DNA fragments containing a chloramphenicol resistance cassette surrounded by sequences homologous to the gene of interest as described in Supplementary Table 3. Colonies were selected on LB agar containing chloramphenicol (20 μg/mL) and transformed again with pCP20 and grown on LB agar containing ampicillin (100 μg/mL) at 30 °C. Finally, clones were grown in LB media at 37 °C. Deletion mutations were confirmed by PCR. All *Salmonella* strains were derived from *S. enterica* serovar Typhimurium strain SL1344[36] and created by allelic exchange as previously described[37].

**Purification of export gate complexes**. FliOPQR or FliOPQR-FlhB were expressed in *E. coli* BL21 (DE3) as a single operon from a pT12 vector (Supplementary Table 3), as described previously[9]. Briefly, cells were grown at 37 °C in TB media containing rhamnose monohydrate (0.1%), collected by spinning at 4000 × *g*, resuspended in TBS (100 mM Tris, 150 mM NaCl, 1 mM EDTA pH 8) and lysed in an EmulsiFlex C5 homogenizer (Avestin). Membranes were prepared from the cleared lysate by ultra-centrifugation at 125,000 × *g* for 3 h. Membranes were solubilized in 1% (w/v) LMNG in TBS and applied to a StrepTrap column (GE Healthcare). The resin was washed in TBS containing 0.01% (w/v) LMNG and proteins were eluted in TBS supplemented with 0.01% (w/v) LMNG and 10 mM desthiobiotin. Intact complexes were separated from aggregate by size-exclusion chromatography in TBS containing 0.01% (w/v) LMNG (S200 10/300 increase or Superose 6 increase, GE Healthcare).

For preparation of FliPQR–FlhB solubilized in the amphipol A8-35, the protein was purified as above using DDM (1% (w/v) for extraction from the membrane and 0.02% (w/v) subsequently) rather than LMNG. Eluate from the StrepTrap column was mixed with amphipol at a ratio of amphipol to protein of 10 : 1. After incubating for 1 h, the sample was dialysed into TBS using a 10,000 MWCO Slide-A-Lyzer device (ThermoFisher Scientific) overnight followed by size-exclusion chromatography on a Superose 6 increase column using TBS as the running buffer.

**Sample preparation for cryo-EM**. Purified complex (3 µl) at 1–3.6 mg/ml were applied to glow-discharged holey carbon-coated grids (Quantifoil 300 mesh, Au R1.2/1.3). Grids were blotted for 3 s at 100% humidity at 22 °C and frozen in liquid ethane using a Vitrobot Mark IV (FEI). For samples solubilized in detergent, blotting was preceded by a wait time of 5–10 s. *V. mimicus* FliPQR was supplemented with 0 mM, 0.05 mM, 0.5 mM or 3 mM fluorinated Fos-Choline prior to grid preparation.

**EM data acquisition and model building**. All data contributing to the final models were collected on a Titan Krios (FEI) operating at 300 kV. All movies were recorded on a K2 Summit detector (Gatan) in counting mode at a sampling of 0.822 Å/pixel, 2.4 e⁻/Å²/frame, 8 s exposure, total dose 48 e⁻/Å², 20 fractions written. Motion correction and dose weighting were performed using MotionCor implemented in Relion 3.0[38] (*V. mimicus* FliPQR–FlhB and *P. savastanoi* FliPQR) or using Simple-unblur[39] (*V. mimicus* FliPQR). Contrast transfer functions were calculated using CTFFIND4[40]. Particles were picked in Simple and processed in Relion 2.0[41] and 3.0[38] as described in Supplementary Figs. 2, 3 and 5.

Atomic models of FliPQR and FlhB were built using Coot[42] and refined in Phenix[43].

**Motility assays**. *E. coli* W strains WL1 or WL2 (Supplementary Table 3) were transformed with plasmids encoding FliOPQR or FlhB containing the mutations to be tested. Saturated overnight cultures (3 µl) were injected into soft agar plates (0.28% agar, 2YT media, containing ampicillin (100 µg/mL) or kanamycin (30 µg/mL) and 0.1% arabinose or 0.5% rhamnose monohydrate as appropriate) and incubated at room temperature.

**E. coli W fT3SS secretion assay**. *E. coli* W strains WL1 or WL2 (Supplementary Table 3) transformed with plasmids encoding FliOPQR or FlhB containing the mutations to be tested were grown up to an OD600 of 1 in 2YT media containing the appropriate antibiotics and 0.1% arabinose or 0.5% rhamnose monohydrate. Cells were pelleted and resuspended in fresh media and grown for another hour. Cells were pelleted again and the supernatant was filtered through a 0.22 µm filter. Proteins were bound to StrataClean beads (Agilent) and the beads were resuspended in SDS-PAGE buffer and run on a 4–20% polyacrylamide gel (Bio-Rad). FliC was detected by immunodetection using an antiserum against *S. typhimurium* FliC.

**Imaging of flagella**. *E. coli* W strains WL1 or WL2 (Supplementary Table 3) transformed with plasmids encoding FliOPQR or FlhB containing the mutations to be tested were grown overnight in 2YT media containing the appropriate antibiotics and 0.1% arabinose or 0.5% rhamnose monohydrate. Culture (10 µl) was applied to glow-discharged carbon-coated grids and stained with 2% uranyl acetate. The grids were imaged using a T12 microscope at 120 kV.

**S. typhimurium vT3SS secretion assay**. Proteins secreted into the culture medium via the vT3SS-1 were analysed as described previously[17]. *S. typhimurium* strains were cultured with low aeration in LB broth supplemented with 0.3 M NaCl at 37 °C for 5 h. Bacterial suspensions were centrifuged at 10,000 × *g* for 2 min and 4 °C to separate whole cells and supernatants. Whole cells were resuspended in SDS-PAGE loading buffer. Supernatants were passed through 0.2 µm pore size filters and supplemented with 0.1 % Na-desoxycholic acid. Proteins in the supernatant were precipitated with 10 % tricholoacetic acid for 30 min at 4 °C and pelleted via centrifugation at 20,000 × *g* for 20 min and 4 °C. Pellets containing precipitated proteins were washed with acetone and resuspended in SDS-PAGE loading buffer. Whole-cell samples and secreted proteins were analysed by SDS-PAGE, western blotting and immunodetection.

**In-vivo photocrosslinking**. In-vivo photocrosslinking was carried out as described previously[19,44] with minor modifications. To enhance expression of vT3SS-1, *S. typhimurium* strains expressed HilA, the master transcriptional regulator of SPI-1 T3SS, from a high copy plasmid under the control of an arabinose-inducible $P_{araBAD}$ promoter[45]. Bacterial cultures were grown in LB broth supplemented with 0.3 M NaCl, 1 mM *p*Bpa and 0.05% arabinose at 37 °C for 5 h. Bacterial cells (5 ODU) were collected and washed once with 5 ml chilled phosphate-buffered saline (PBS) to remove residual media. Bacteria were pelleted by centrifugation at 4000 × *g* for 3 min and 4 °C, and afterwards resuspended in 1 ml of chilled PBS. Bacterial suspensions were transferred into six-well cell culture dishes and irradiated for 30 min with UV light ($\lambda = 365$ nm) on a UV transilluminator table (UV). Subsequently, bacterial cells were pelleted by centrifugation at 10,000 × *g* for 2 min and 4 °C. Samples were stored at −20 °C until use.

**Crude membrane preparation**. Crude membranes were purified following the published protocol[19]. Bacterial cells (5 ODU) were resuspended in 750 µl lysis buffer (50 mM triethanolamine pH 7.5, 250 mM sucrose, 1 mM EDTA, 1 mM MgCl₂, 10 µg/ml DNAse, 2 µg/ml lysozyme, 1:100 protease inhibitor cocktail) and incubated on ice for 30 min. Cell slurries were lysed via continuous bead milling. Intact cells, beads and debris were removed by centrifugation for 10 min at 10,000 × *g* and 4 °C. Supernatants were centrifuged for 50 min at 52,000 r.p.m. and 4 °C in a Beckman TLA 55 rotor to pellet bacterial membranes. Pellets containing crude membranes were store at −20 °C until use. Samples were analysed by SDS-PAGE, western blotting and immunodetection.

**Western blotting and immunodetection**. Samples were loaded onto SERVAGel™ TG PRiME 8–16% precast gels and transferred on polyvinylidene difluoride (PVDF) membranes (Bio-RAD). Proteins were detected with primary antibodies anti-*St₁*SctP (InvJ)[17] (1 : 2000) or M2 anti-FLAG (1 : 10,000) (Sigma-Aldrich, F3165). Secondary antibodies (ThermoFisher, SA5-35571) were goat anti-mouse IgG Dylight 800 conjugate (1 : 5000). Scanning of the PVDF membranes and image analysis was performed with a Li-Cor Odyssey system and Image Studio 3.1 (Li-Cor).

**Reporting summary**. Further information on research design is available in the Nature Research Reporting Summary linked to this article.

## Data availability
Cryo-EM volumes and atomic models have been deposited to the EMDB (accession codes EMD-10095, EMD-10096, EMD-10093 and EMD-10653) and PDB (accession codes 6S3R, 6S3S and 6S3L), respectively. The source data underlying Figs. 1b, c, 2a,3d and 4b–d and Supplementary Figs. 2b, 3b, 4, 5c, 11, 12 and 13 are provided as a Source Data file. Other data are available from the corresponding author upon reasonable request.

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

## Acknowledgements

We thank E. Johnson and A. Costin of the Central Oxford Structural Microscopy and Imaging Centre for assistance with data collection. H. Elmlund (Monash) is thanked for assistance with access to SIMPLE code ahead of release. We thank G. Fraser, Cambridge, for providing materials. The Central Oxford Structural Microscopy and Imaging Centre is supported by the Wellcome Trust (201536), the EPA Cephalosporin Trust, the Wolfson Foundation, and a Royal Society/Wolfson Foundation Laboratory Refurbishment Grant (WL160052). Work performed in the lab of S.M.L. was supported by a Wellcome Trust Investigator Award (100298) and an MRC Program Grant (M011984). L.K. is a Wellcome Trust PhD student (109136). Work in the laboratory of S.W. related to this project was funded by the German Center for Infection Research (DZIF), project TTU06.801/808.

## Author contributions

L.K. performed experiments, did strain and plasmid construction, complex purification, native mass spectrometry, cryo-EM grid optimization, cryo-EM data analysis, and model building and analysis. J.D. performed experiments, cryo-EM grid optimization and data collection. J.J.E.C. performed data analysis. J.F. performed experiments and complex purification. A.Z., S.B. and R.D. generated pBpa mutants, performed crosslinking experiments and secretion assays, and analysed data. S.W. designed injectisome functional experiments and analysed data. S.J. and S.M.L. supervised experimental work and wrote the first draft of the paper with L.K. All authors contributed to and commented on the final manuscript.

## Competing interests

The authors declare no competing interests.
