## [Peer Review File · Nature Communications]

Reviewers' comments:

Reviewer #1 (Remarks to the Author):

The flagellum and related injectisome are macromolecular machines of remarkable complexity. The assembly of both the flagellum and injectisome relies on protein export via a homologous type-III secretion system (T3SS). The core export gate complex of the T3SS is made of FliP, FliQ, FliR, FlhB and FlhA (SctRSTUV in the injectisome).

However, how the T3SS functions on a molecular level remains poorly understood. The authors previously reported the fascinating structure of the T3SS core complex made of FliP5Q4R1 (Kuhlen et al. *Nature Structural & Molecular Biology* 25, 583-590 (2018)).

In the present manuscript, Kuhlen and colleagues report the structure the core export gate complex that additionally includes FlhB. FlhB is an essential component of the T3SS and involved in switching from early to late substrate secretion mode.

The structure reported by Kuhlen et al. surprisingly wraps around the gate of the FliPQR secretion pore, suggesting that FlhB is involved in opening of the export gate.

The manuscript is well written and provides the next missing piece in the author's endeavor to solve the complete structure of the type-III export apparatus. Accordingly, the present report is well-suited for the broad readership of *Nature Communications* and I have only few comments that might help the authors to improve the clarity of their manuscript.

1) The authors solved the structure of the export gate complex of the flagellar T3SS, performed functional assays using mutants of the flagellar T3SS, however, used the injectisome T3SS homolog for their *in vivo* photocrosslinking experiments. Accordingly, the manuscript is a mix of flagellar (FliPQR FlhBA) and injectisome (SctRSTUV) nomenclature, which unfortunately, is mildly confusing. The authors should consider to at least harmonize the amino acid numbering throughout their figures e.g. using the numbering of the *V. mimicus* FlhB homolog in order to facilitate interpretation of the reported data.

2) Fig. 1c: SctT-SctU crosslinks are also observed in the absence of UV irradiation?

3) Fig. 2: It might be useful to discuss the existence of a fliR-flhB gene fusion in *Clostridium* in light of their FlhB structure.

4) Fig. 4a: The motility soft agar assays of FliQ and FlhB point mutants are hard to interpret. The authors should highlight the fact that motility in soft agar plates is dependent on several factors, including growth, chemotaxis and flagella assembly.

5) Fig. 4a: It is unclear if the generated point mutations result in a defect in export apparatus assembly or directly affect protein secretion via the flagellar T3SS.

6) Fig. 4a: The introduction of charged residues in FliQ completely abolished motility, presumably because the FliQ-FlhB interaction is disrupted. It might be interesting to introduce an additional mutation with the opposite charge in the interacting residue of FlhB in order to restore the presumed interaction defect.

7) Fig. 4a: Can the authors comment on why the flhB deletion mutant is only poorly complemented by wildtype FlhB?

8) Fig. 4a: The authors propose that a linker between TM2 and 3 of FlhB has a role in opening of the secretion gate. Consistently, they show that deletions of six or more residues in the FlhB linker reduces motility. However, such deletions might also change the placement of FlhB_{TM} and thereby disrupt the FlhB-FliQ interaction. If the FlhB loop is directly involved in controlling the opening state of the export gate, it might also not tolerate insertions?

9) The genetic nomenclature is inconsistent throughout the manuscript and figures. For details see Demerec et al. *Genetics* 54, 61 (1966).

10) Suppl. Figure 11 legend: reference error.

11) Suppl. Figure 12: Could the unidentified SctU crosslinks be crosslinks to substrates? Unfortunately, the authors do not discuss these crosslinks in the manuscript.

12) Discussion: The authors propose that the present FlhB structure explains the previously reported FlhB_P28T mutation, which has been shown to allow T3SS function in the absence of the ATPase complex FliHI. It is unclear if the authors propose a model, in which (in the native T3SS), the ATPase complex (presumably the stalk protein FliJ) directly interacts with FlhB in order to facilitate the opening of the export gate. Would this be possible simply considering the rather long distance between the ATPase complex located at the base of the C-ring and the export gate within the basal body?

13) Discussion: The authors propose a model in which cycles of ATP hydrolysis via the ATPase complex FliHIJ induce conformational changes in the export gate (via FliJ-FlhA-FlhB?), which results in gate opening. In this respect it might be useful to discuss previous findings showing that secretion via the flagellar T3SS is possible in the absence of the ATPase complex or in mutants strongly impaired in ATPase activity. In light of the present FlhB structure, these reports support a model where the ATPase is only required for the initial gate opening (which, apparently, can also occur spontaneously). It might be reasonable to assume that, once opened, the export gate remains open as long as substrate proteins are transported.

Reviewer #2 (Remarks to the Author):

The type III secretion system (T3SS) of pathogenic bacteria is utilized not only for direct injection of effector proteins into host cell for bacterial infection but also for construction of the bacterial flagellum, which is responsible for bacterial motility. A protein export apparatus of the T3SS is composed of a transmembrane export gate complex and an associated ATPase ring complex. A core complex of the flagellar export gate complex is composed of five copies of FliP, a single copy of FliR and four copies of FliQ, and forms a right-handed helical structure inside the basal body MS ring to act as a protein translocation channel. Two transmembrane proteins, FlhA and FlhB, coordinates flagellar protein export with assembly by ordered export of flagellar proteins to parallel with their order of assembly. Fukumura et al. have shown that FlhA and FlhB are associated with the FliPQR complex but it remains unknown how. In this study, Kuhlen et al. have reported a cryoEM structure of the FliPQR-FlhB complex at a resolution of 3.2 Å. Structure-based mutational analyses have shown that a long loop between helices 2 and 3 (residues 110–139) is responsible for export gating function. All experiments are carefully and well designed. Also, all data are clearly presented and well documented. This research article would be of interest to general readership, providing important advancements in our knowledge on the protein export mechanism of the T3SS. However, this reviewer has several comments.

1. The authors solved the flagellar FliPQR-FlhB structure at 3.2 Å by cryoEM and single particle image analysis. Based on this structure, they carried out functional analysis of the *Salmonella* SPI-1 T3SS rather than the flagellar T3SS. This reviewer strongly suggest that the authors should show photo-crosslinking data of the *Vibrio* flagellar T3SS based on their structure to clarify their cyroEM structure (Figures 1b, 4b and 4d). The authors should perform secretion assays of flagellar proteins such as the hook protein FlgE and the filament protein, FliC to clarify that the loop connecting helices 2 and 3 are involved in export gating function.

2. P2, line 15: The authors should cite the following two papers in addition to reference 6.

Kawamoto, A. et al. Common and distinct structural features of Salmonella injectisome and flagellar basal body. *Sci. Rep.* 3, 3369 (2013).

Terahara, N. et al. Insight into structural remodeling of the FlhA ring responsible for bacterial flagellar type III protein export. *Sci. Adv.* 4, eaao7054 (2018).

3. P12, lines 2 – 4: There is no evidence and so this sentence is too speculative.

4. P12, line 15: The reference 23 showed that motility of the Salmonella flhB(D208A) mutant is restored to nearly wild-type level, suggesting that Asp-208 may be involved in the interaction with FlhA. The authors should describe this point in Discussion.

5. P13, lines 1–4: The authors should cite the following two papers because these two papers provide direct evidences an interaction between FliJ and FlhA.

Minamino, T., Morimoto, Y.V., Hara, N. & Namba, K. An energy transduction mechanism used in bacterial flagellar type III protein export. *Nat. Commun.* 2, 475 (2011).

Ibuki, T. et al. Interaction between FliJ and FlhA, components of the bacterial flagellar type III export apparatus. *J. Bacteriol.* 195, 466–473 (2013).

Reviewer #3 (Remarks to the Author):

The manuscript studied a complex structure of FliPRQ and FlhB in T3SS. The structure of FlhB in the complex is new, and shows binding near export gate of FliPRQ complex. This finding is interesting and important to understand the role of FlhB in the secretion by T3SS. However, several concerns must be addressed before publication.

1) The loops in FlhB present near the export gate, and its interaction with the export gate was observed and validated. And it is proposed that the loops of FlhB, such as FlhBL, are involved in the gate opening. Mutations on FlhBL can severely reduce the motility but not affect binding of FlhB to FliPQR. Is it possible that FlhBL has important interactions with the substrate, and hence the mutations influenced something for the interaction with substrate and reduced the motility, rather than only the gate opening?

2) FlhB can bind on the same position as FliQ5. What is the affinity of this binding, stronger than FliQ5? Is it possible that FliQ5 competitively binds on FliPQR complex in vivo?

3) line 28 page 8, "this loop, c which ..." what is c ?

4) It is better to label Helix 1, 2, 3, and 4 in Fig3b and c, as well as FlhBL.

5) line 27 page 3, "using cryo-EM and single particle analysis" should be "using single-particle cryo-EM analysis"

6) in Fig 1, the position of membrane should be marked.

Response to Referees

Reviewer #1 (Remarks to the Author):

The flagellum and related injectisome are macromolecular machines of remarkable complexity. The assembly of both the flagellum and injectisome relies on protein export via a homologous type-III secretion system (T3SS). The core export gate complex of the T3SS is made of FlhP, FliQ, FliR, FlhB and FlhA (SctRSTUV in the injectisome).

However, how the T3SS functions on a molecular level remains poorly understood. The authors previously reported the fascinating structure of the T3SS core complex made of FlhP5Q4R1 (Kuhlen et al. *Nature Structural & Molecular Biology* 25, 583-590 (2018)).

In the present manuscript, Kuhlen and colleagues report the structure the core export gate complex that additionally includes FlhB. FlhB is an essential component of the T3SS and involved in switching from early to late substrate secretion mode.

The structure reported by Kuhlen et al. surprisingly wraps around the gate of the FlhPQR secretion pore, suggesting that FlhB is involved in opening of the export gate.

The manuscript is well written and provides the next missing piece in the author's endeavor to solve the complete structure of the type-III export apparatus.

Accordingly, the present report is well-suited for the broad readership of *Nature Communications* and I have only few comments that might help the authors to improve the clarity of their manuscript.

We are pleased the reviewer shares our conviction that the work is likely of interest for the broad readership of *Nature Communications* and give point-by-point responses to their comments below:

The authors solved the structure of the export gate complex of the flagellar T3SS, performed functional assays using mutants of the flagellar T3SS, however, used the injectisome T3SS homolog for their *in vivo* photocrosslinking experiments. Accordingly, the manuscript is a mix of flagellar (FlhPQR FlhBA) and injectisome (SctRSTUV) nomenclature, which unfortunately, is mildly confusing. The authors should consider to at least harmonize the amino acid numbering throughout their figures e.g. using the numbering of the *V. mimicus* FlhB homolog in order to facilitate interpretation of the reported data.

This is a helpful suggestion and we have implemented throughout our manuscript using the *V. mimicus* as the point of reference

2) Fig. 1c: SctT-SctU crosslinks are also observed in the absence of UV irradiation?

Close inspection of this band reveals that this is an unspecific background band rather than a SctT-SctU cross-linked species.

3) Fig. 2: It might be useful to discuss the existence of a *fliR-flhB* gene fusion in *Clostridium* in light of their FlhB structure.

We have discussed interpretation of these earlier observations starting on page 6 line 21.

“...Interestingly, both termini of FlhB_{TM} are cytoplasmic, while the C-terminus of FliR is periplasmic. Perplexingly the observation that an engineered fusion of FliR-FlhB in *Salmonella*(20) weakly complements a double *fliR/flhB* knockout, in conjunction with the existence of a natural FliR-FlhB fusion in *Clostridium*, led to previous suggestions that the C-terminus of FliR and the N-terminus of FlhB are either both cytoplasmic or both periplasmic. In light of our structure we would either have to accept that the N-terminus of FlhB (residues 1-28 that are not observed in our structure) can wrap over the surface of the FlhPQR-FlhB complex to reach towards the C-terminus of FliR, or that the low level of activity seen in the engineered *Salmonella* fusion (20) and activity in the native *Clostridial* species relies on proteolytic separation of the two proteins.

4) Fig. 4a: The motility soft agar assays of FliQ and FlhB point mutants are hard to interpret. The authors should highlight the fact that motility in soft agar plates is dependent on several factors, including growth, chemotaxis and flagella assembly.

We have added an explanation of how multiple linked factors contribute to the presence or absence of motility on page 10 starting at line 10.

“...Our motility assays (Fig. 4a) could be influenced by multiple factors including growth of the cells, assembly of flagella, chemotaxis and secretion. In order to assay secretion more directly, we tested secretion of the early secretion substrate SctP in the *pBpa* mutants...”

5) Fig. 4a: It is unclear if the generated point mutations result in a defect in export apparatus assembly or directly affect protein secretion via the flagellar T3SS.

We previously addressed this by demonstrating that the mutations associated with a complete loss of motility did not disrupt complex assembly by purifying the mutant forms of the complex (previously in supplementary figure 11, now supplementary figure 13). We have now added equivalent data (supplementary figure 13) for the other functionally altered mutants demonstrating that all the mutants can still form a stable, purifiable complex. We have also tested mutant secretion of FliC (Supplementary Fig. 11) and inspected the bacteria for presence/absence of flagella (Supplementary Fig. 12). We have altered the wording to emphasise our interpretation of these data.

'...Mutations in either the conserved hydrophobic residues of FlhB_L that insert between the FliQ subunits (Fig. 3f) or the highly conserved loop of FliQ severely reduced motility (Fig. 4a), protein secretion (Supplementary Fig.11) and formation of flagella (Supplementary Fig. 12) without affecting export apparatus assembly (Supplementary Fig. 13)...

6) Fig. 4a: The introduction of charged residues in FliQ completely abolished motility, presumably because the FliQ-FlhB interaction is disrupted. It might be interesting to introduce an additional mutation with the opposite charge in the interacting residue of FlhB in order to restore the presumed interaction defect.

While the suggested experiment would be interesting, the stoichiometry of the complex means that each point mutation in FliQ contacts four completely different regions of FlhB and therefore the construction and interpretation of such mutants would not be trivial, especially as we do not have a suitable flhB null mutant background in which to perform such studies. Furthermore, introduction of the charged residues in FliQ does not impact on export apparatus assembly, as now demonstrated in supplementary figure 13 and likely impacts more subtly on local conformation and ability to couple conformational changes between the proteins.

7) Fig. 4a: Can the authors comment on why the flhB deletion mutant is only poorly complemented by wildtype FlhB?

The flhB gene is the first in the flhBAE operon and it is commonly observed that deletion of the first gene in an operon leads to lower expression of the downstream genes. We assume this explains the not-wild type level of complementation in this case.

8) Fig. 4a: The authors propose that a linker between TM2 and 3 of FlhB has a role in opening of the secretion gate. Consistently, they show that deletions of six or more residues in the FlhB linker reduces motility. However, such deletions might also change the placement of FlhB_{TM} and thereby disrupt the FlhB-FliQ interaction. If the FlhB loop is directly involved in controlling the opening state of the export gate, it might also not tolerate insertions?

This is an interesting idea and we have therefore constructed a new mutant containing a six residue insertion in this loop and looked at the impact on function (Fig. 4). Interestingly this duplication was less motile than both the wild type and a three residue deletion in the same loop, but was better tolerated than the six residue deletion. This may be because looping out of the additional residues allows many of the contacts between FlhB_L and FliQ to be maintained in both closed and putative open states.

"...If this interaction is maintained in the open state of the export gate, FlhB_L would have to adopt a more extended conformation, requiring a minimum number of residues in the loop. Consistent with this idea, deletions of six or more residues in FlhB_L led to loss of motility whilst duplication of six residues in FlhB_L was rather better tolerated, leading only to a reduction in motility (Fig. 4a)..."

9) The genetic nomenclature is inconsistent throughout the manuscript and figures. For details see Demerec et al. Genetics 54, 61 (1966).

We apologise for the inconsistencies and have tried to adhere more consistently to established standards.

10) Suppl. Figure 11 legend: reference error.

Apologies – now fixed.

11) Suppl. Figure 12: Could the unidentified SctU crosslinks be crosslinks to substrates? Unfortunately, the authors do not discuss these crosslinks in the manuscript.

Interpretation of these cross-links is complex and we do not have a definitive interpretation however we are confident that these do not reflect cross-links to substrate since we observe that they occur in non-secreting mutants (like SctN_{K165E}, data not shown). We would speculate that these crosslinks may relate to interactions of the SctU protomer, of SctU in an assembly intermediate, or of unproductive assembly/malfolded SctU. In support of these relating to some assembly intermediate or dead-end, mis-assembled complex we note that the prominent SctU_{A109X} crosslink is never seen in SctU incorporated in the full assembly but only occurs with SctU assembled in the SctRSTU assembly intermediate (2D BN PAGE, data not shown). Since these observations do not directly inform interpretation of the structure presented here, we have not added this additional discussion in the revised text.

12) Discussion: The authors propose that the present FlhB structure explains the previously reported FlhB_{P28T} mutation, which has been shown to allow T3SS function in the absence of the ATPase complex FliH. It is unclear if the authors propose a model, in which (in the native T3SS), the ATPase complex (presumably the stalk protein FliJ) directly interacts with FlhB in order to facilitate the opening of the export gate. Would this be possible simply considering the rather long distance between the ATPase complex located at the base of the C-ring and the export gate within the basal body?

We apologise that our logic was unclear and have re-written to hopefully clarify our thinking (page 13 starting at line 3)

“...This functional link between the ATPase and the gating mechanism, in conjunction with a host of other mutational data in FlhA and FlhB, suggests that cycles of ATP hydrolysis may induce conformational changes in the export gate. As the distance between the export gate and ATPase is large, these conformational changes would presumably be mediated via the FlhA ring that is positioned between them and has been shown to interact with the FliJ stalk of the ATPase complex(32, 33). Since low levels of secretion are possible even in the absence of the ATPase complex or in mutants impaired in ATPase activity (22), low level, spontaneous, opening of the export gate complex must also be possible. Whether the gate subsequently stays open, for example due to the continued presence of substrate in the channel (29), or whether it opens and closes as one substrate molecule after the other is injected into the growing filament, as might be expected from the injection-diffusion model (34), is unknown...”

13) Discussion: The authors propose a model in which cycles of ATP hydrolysis via the ATPase complex FliHJ induce conformational changes in the export gate (via FliJ-FlhA-FlhB?), which results in gate opening. In this respect it might be useful to discuss previous findings showing that secretion via the flagellar T3SS is possible in the absence of the ATPase complex or in mutants strongly impaired in ATPase activity. In light of the present FlhB structure, these reports support a model where the ATPase is only required for the initial gate opening (which, apparently, can also occur spontaneously). It might be reasonable to assume that, once opened, the export gate remains open as long as substrate proteins are transported.

We have added a discussion about these data on page 13.

“...Since low levels of secretion are possible even in the absence of the ATPase complex or in mutants impaired in ATPase activity (22), low level, spontaneous, opening of the export gate complex must also be possible...”

Reviewer #2 (Remarks to the Author):

The type III secretion system (T3SS) of pathogenic bacteria is utilized not only for direct injection of effector proteins into host cell for bacterial infection but also for construction of the bacterial flagellum, which is responsible for bacterial motility. A protein export apparatus of the T3SS is composed of a transmembrane export gate complex and an associated ATPase ring complex. A core complex of the flagellar export gate complex is composed of five copies of FliP, a single copy of FliR and four copies of FliQ, and forms a right-handed helical structure inside the basal body MS ring to act as a protein translocation channel. Two transmembrane proteins, FlhA and FlhB, coordinates flagellar protein export with assembly by ordered export of flagellar proteins to parallel with their order of assembly. Fukumura et al. have shown that FlhA and FlhB are associated with the FliPQR complex but it remains unknown how. In this study, Kuhlén et al. have reported a cryoEM structure of the FliPQR-FlhB complex at a resolution of 3.2 Å. Structure-based mutational analyses have shown that a long loop between helices 2 and 3 (residues 110–139) is responsible for export gating function. All experiments are carefully and well designed. Also, all data are clearly presented and well documented. This research article would be of interest to general readership, providing important advancements in our knowledge on the protein export mechanism of the T3SS.

We thank the reviewer for their thoughtful appreciation of our work.

However, this reviewer has several comments.

We address these below

The authors solved the flagellar FliPQR-FlhB structure at 3.2 Å by cryoEM and single particle image analysis. Based on this structure, they carried out functional analysis of the Salmonella SPI-1 T3SS rather than the flagellar T3SS. This reviewer strongly suggest that the authors should show photo-crosslinking data of the Vibrio flagellar T3SS based on their structure to clarify their cryoEM structure (Figures 1b, 4b and 4d). The authors should perform secretion assays of flagellar proteins such as the hook protein FlgE and the filament protein, FliC to clarify that the loop connecting helices 2 and 3 are involved in export gating function.

The high level of conservation between the secretion and flagellar type three systems means that we are confident in drawing conclusions based on experiments across the different systems. The data have therefore been collected using strains that were readily available to us, however to address this point and formally demonstrate that the mutant complexes (1) assemble and (2) prevent secretion of substrates we present the data in supplementary figures 11 and 12 demonstrating that the complexes can still be assembled in the with the mutations present, but that no flagella are assembled. We have also carried out FliC secretion assays using the mutant strains and demonstrated a clear correlation between lack of motility and lack of FliC secretion (supplementary figure 11)

‘...Mutations in either the conserved hydrophobic residues of FlhB_L that insert between the FliQ subunits (Fig. 3f) or the highly conserved loop of FliQ severely reduced motility (Fig. 4a), protein secretion (Supplementary Fig.11) and formation of flagella (Supplementary Fig. 12) without affecting export apparatus assembly (Supplementary Fig. 13)...’

2. P2, line 15: The authors should cite the following two papers in addition to reference 6.

Kawamoto, A. et al. Common and distinct structural features of Salmonella injectisome and flagellar basal body. Sci. Rep. 3, 3369 (2013).

Terahara, N. et al. Insight into structural remodeling of the FlhA ring responsible for bacterial flagellar type III protein export. Sci. Adv. 4, eaao7054 (2018).

We have added these references as #7 and #8 respectively

3. P12, lines 2 – 4: *There is no evidence and so this sentence is too speculative.*

Whilst we appreciate that the evidence is indirect and this model remains to be validated further, the loss of motility in mutants in which the complex can still assemble but which are designed to disrupt the details of the interaction between FliQ and FlhB suggest that the details of this interaction, as seen in our structure, are required for secretion. Since FliPQR will need to open to allow passage of substrates, FlhB_L must either dissociate and be displaced from the secretion pathway or extend to allow the opening whilst still interacting the FliQ (as discussed on p9). Our interpretation of both the point mutations and the mutants that alter FlhB_L length, is that extension whilst interacting with FliQ is the more likely scenario. We have checked our language in discussion of this throughout and hope that the use of modifiers such as “suggesting” and “propose” allow the reader to understand both our logic and the idea that this is in no way a proven fact. However, we feel it is important to communicate what we feel is the most likely molecular model for gating.

4. P12, line 15: *The reference 23 showed that motility of the Salmonella flhB(D208A) mutant is restored to nearly wild-type level, suggesting that Asp-208 may be involved in the interaction with FlhA. The authors should describe this point in Discussion.*

This is an interesting point. We have added a sentence noting this on p12

“... Interestingly, this network of charged residues in FlhB includes D208. The D208A mutation disrupts motility, but this defect can be rescued by overexpression of FlhA(28), reinforcing the functional link between the charge network in FlhB and FlhA...”

5. P13, lines 1–4: *The authors should cite the following two papers because these two papers provide direct evidences an interaction between FliJ and FlhA. Minamino, T., Morimoto, Y.V., Hara, N. & Namba, K. An energy transduction mechanism used in bacterial flagellar type III protein export. Nat. Commun. 2, 475 (2011).*

Ibuki, T. et al. Interaction between FliJ and FlhA, components of the bacterial flagellar type III export apparatus. J. Bacteriol. 195, 466–473 (2013).

We have added these references as #32 and #33 respectively

Reviewer #3 (Remarks to the Author):

The manuscript studied a complex structure of FliPQR and FlhB in T3SS. The structure of FlhB in the complex is new, and shows binding near export gate of FliPQR complex. This finding is interesting and important to understand the role of FlhB in the secretion by T3SS.

We thank the reviewer for their appreciation of our work

However, several concerns must be addressed before publication.

We address these point-by-point below

1) *The loops in FlhB present near the export gate, and its interaction with the export gate was observed and validated. And it is proposed that the loops of FlhB, such as FlhBL, are involved in the gate opening. Mutations on FlhBL can severely reduce the motility but not affect binding of FlhB to FliPQR. Is it possible that FlhBL has important interactions with the substrate, and hence the mutations influenced something for the interaction with substrate and reduced the motility, rather than only the gate opening?*

This is, of course, a possibility. However, since the strongest phenotypes are seen in mutations that occur in the contacts to the FliQ subunits (A41, T42, L128, M137) rather than those directed into the lumen of the loop we find this a less probable an explanation.

2. *FlhB can bind on the same position as FliQ5. What is the affinity of this binding, stronger than FliQ5? Is it possible that FliQ5 competitively binds on FliPQR complex in vivo?*

Whilst we cannot rule out that a fifth FliQ competes with FlhB for this binding site during assembly, we are not aware of a method that would allow us to measure affinity of binding between integral membrane proteins that only assemble during co-expression so it is therefore not possible to directly address the question. Since we have only seen occupancy of FliQ5 in the single species where we have expressed FliPQR in the absence of FlhB and since, even in this case, the binding is sub stoichiometric (Kuhlen et al. Nature Structural & Molecular Biology 25, 583-590 (2018)) we assume that a fifth FliQ is only weakly bound and where FlhB is present it will always outcompete an additional copy of FliQ. Expression of FliOPQR and FlhB from the same operon may facilitate correct assembly in vivo and we note that the recent structure of the export gate *in situ*, sees only four copies of the FliQ homologue (Hu et al. Nature Microbiology 4, 2010-2019 (2019)).

3. *line 28 page 8, “this loop, c which ...” what is c ?*

We apologise for this error - the character c was entirely spurious and has been removed

4) It is better to label Helix 1,2, 3, and 4 in Fig3b and c, as well as FlhBL.

We have added these labels as suggested

5) line 27 page 3, "using cryo-EM and single particle analysis" should be "using single-particle cryo-EM analysis"

Altered as requested

6) in Fig 1, the position of membrane should be marked.

We respectfully disagree with the reviewer as the true 'position' of the membrane is a non-trivial thing to annotate. Does the reviewer want us to indicate the position of the detergent micelle in our structures? This presumably reflects how the complexes sit in the bacterial inner membrane *prior* to assembly of the flagellar basal body. We have chosen however not to indicate this, as the position so indicated would only be of relevance to the basal-body assembly intermediate. Whilst we are confident that the plane of the inner membrane would lie below the assembled export apparatus, our current lack of knowledge of the structure of the transmembrane portion of FlhA means that we cannot, with any confidence, understand how much lipid bilayer exists within the flagellar basal body. In fact, we find it likely, that the complex we here study does not make any contacts with bilayer but rather positions the C-terminal portion of FlhB to enter the cytoplasm via a proteinaceous channel in FlhA. For all these complex reasons we do not wish to potentially mislead the reader by indicating a simple location of the membrane in this panel.

REVIEWERS' COMMENTS:

Reviewer #1 (Remarks to the Author):

Kuhlen et al. substantially revised their manuscript and appropriately addressed my concerns. I do not have any further comments and recommend publication of the manuscript.